# DCAFuse: Dual-Branch Diffusion-CNN Complementary Feature Aggregation Network for Multi-Modality Image Fusion

## ABSTRACT

Multi-modality image fusion (MMIF) aims to integrate the complementary features of source images into the fused image, including target saliency and texture specifics. Recently, image fusion methods leveraging diffusion models have demonstrated commendable results. Despite their strengths, diffusion models reduce the capability to perceive local features. Additionally, their inherent working mechanism, introducing noise to the inputs, consequently leads to a loss of original information. To overcome this problem, we propose a novel Diffusion-CNN feature Aggregation Fusion (DCAFuse) network that can extract complementary features from the dual branches and aggregate them effectively. Specifically, we utilize the denoising diffusion probabilistic model (DDPM) in the diffusion-based branch to construct global information, and multi-scale convolutional kernels in the CNN-based branch to extract local detailed features. Afterward, we design a novel complementary feature aggregation module (CFAM). By constructing coordinate attention maps for the concatenated features, CFAM captures long-range dependencies in both horizontal and vertical directions, thereby dynamically guiding the aggregation weights of branches. In addition, to further improve the complementarity of dual-branch features, we introduce a novel loss function based on cosine similarity and a unique denoising timestep selection strategy. Extensive experimental results show that our proposed DCAFuse outperforms other state-of-the-art methods in multiple image fusion tasks, including infrared and visible image fusion (IVF) and medical image fusion (MIF). The source code will be publicly available at https://xxx/xxx/xxx.

## CCS CONCEPTS

• **Computing methodologies** → *Image processing*.

## KEYWORDS

Multi-modality image fusion, diffusion model, feature aggregation

## 1 INTRODUCTION

Multi-modality image fusion (MMIF) generates information-rich fused images by integrating the complementary features of different categories of source images [5, 83, 86, 91]. Infrared and Visible Fusion (IVF) and Medical Image Fusion (MIF) are typical tasks in MMIF. Specifically, IVF aims to integrate the saliency information in

*ACM MM, 2024, Melbourne, Australia*
© 2024 Copyright held by the owner/author(s). Publication rights licensed to ACM.
ACM ISBN 978-x-xxxx-xxxx-x/YY/MM
https://doi.org/10.1145/nnnnnnn.nnnnnnn

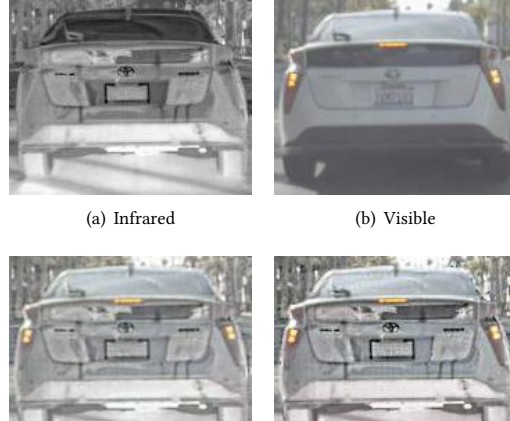

(a) Infrared     (b) Visible

(c) Dif-Fusion (TIP'23) [79]     (d) DCAFuse (Ours)

**Figure 1: Visual comparisons of fusion results between (c) the conventional diffusion-based method Dif-Fusion [79] and (d) the proposed DCAFuse. Notably, DCAFuse showcases clearer contour details and improved contrast compared to the existing method.**

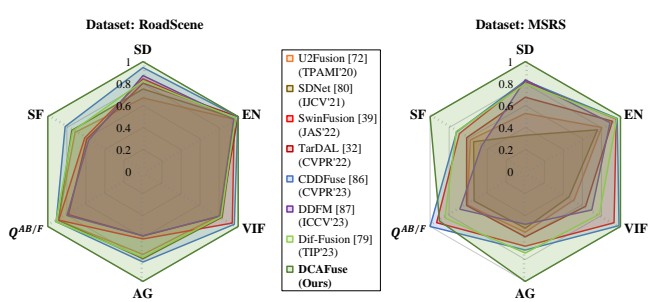

**Figure 2: Fusion evaluation metrics comparison on Road-Scene [73] and MSRS [57] datasets.**

infrared images and the texture details in visible images to produce results with prominent targets and clear backgrounds [27, 38, 43], which are widely used in fields such as autonomous driving [8, 77], drone nighttime monitoring [53, 69], video surveillance [10, 25], etc. On the other hand, MIF combines images obtained from various medical imaging modalities [22, 31], such as MRI and CT, to assist in medical diagnosis and treatment [17].

Numerous deep learning-based approaches have been developed to tackle the challenges of MMIF, mainly encompassing methods founded on Convolutional Neural Networks (CNNs) and generative models [24, 54, 81]. CNN-based techniques, restrained to a limited

receptive field for feature extraction, fall short of sufficiently capturing cross-modality information and long-range dependencies, thereby undermining information fidelity [26, 27, 58]. Although Generative Adversarial Networks (GANs), renowned generative models, capably model source image distribution in alignment with MMIF requirements [32, 42], their reliance on adversarial interactions in the fusion process fosters modality imbalance and convergence issues, hence diminishing fusion quality [40, 41]. Recent developments in diffusion models, such as Denoising Diffusion Probabilistic Models (DDPM) [18], exhibit outstanding global information modeling capabilities [20, 68] and outperform GANs in image generation quality [9, 15]. Therefore, numerous studies attempt to apply diffusion models to visual tasks such as MMIF [6, 64, 87]. The common technique involves initially infusing noise into source images, followed by extraction of latent features during the denoising process for fusion, leading to the production of visually impressive fused images [3, 4, 79].

However, existing diffusion-based fusion methods often compromise critical texture details in the source images [79, 87]. As illustrated in Fig. 1(c), critical features such as the car's logo and contours in the Dif-Fusion fused image appear blurred. This deficiency stems from inherent limitations in existing diffusion-based MMIF methods: (i) Inability to extract local detailed features. Although latent features extracted from the denoising network can represent global information, they lack localized perception capabilities like CNNs [12, 65]. (ii) Inherent degradation of original information. The working mechanism of diffusion-based fusion methods necessitates the introduction of noise to source images [18, 23, 52], inevitably leading to the loss of original information. (iii) Insufficient exploration of effective timestep combinations. Intermediate features at diverse denoising timesteps exhibit distinct regional perceptions [13], requiring a tailored design of selection strategy for effective fusion. However, comprehensive explorations of this aspect are lacking in existing works [79, 87].

To address the drawbacks mentioned above, we propose a novel dual-branch Diffusion-CNN feature Aggregation Fusion (DCAFuse) network, capable of extracting complementary features in terms of perceptual range through CNN-based and diffusion-based branches, and effectively aggregating the features based on their long-range dependencies. Specifically, in the diffusion-based branch, we extract the intermediate features at multiple timesteps of the DDPM to construct global information. In the CNN-based branch, multi-scale convolutional kernels are utilized to extract local detailed features. Afterward, we propose a novel complementary feature aggregation module (CFAM) to effectively aggregate the concatenated features. By generating coordinate-aware attention maps, CFAM captures the long-range dependencies in both horizontal and vertical directions [19], thus dynamically guiding the aggregating weights, and then the aggregated features are fed into the fusion head to output. Moreover, to further enhance the complementarity of the features extracted from each branch, we introduce a cosine divergence loss function and an innovative denoising timestep selection strategy different from the existing methods. Fig. 1(d) shows the fusion result of DCAFuse, which exhibits much clearer contour details and better contrast than the existing method. When compared with state-of-the-art methods in various evaluation metrics, our method

also achieves leading performance, as shown in Fig. 2. Overall, our contributions are summarized as follows:

- We propose DCAFuse, a dual-branch diffusion-CNN framework for multi-modality image fusion, leveraging both the global information modeling capability of DDPM and the local detailed feature extraction capability of multi-scale convolutional kernels.
- We propose a novel Complementary Feature Aggregation Module (CFAM) based on the coordinate attention mechanism. It can perceive the long-range dependencies of the dual-branch features in both horizontal and vertical directions, thus generating coordinate-aware attention maps to dynamically guide feature aggregation.
- We introduce a cosine divergence loss function and a unique denoising timestep selection strategy, effectively enhancing the complementarity of the features extracted from each branch.
- Experiments on multi-modality image fusion demonstrate the superiority of our DCAFuse, improving the average gradient (AG) and spatial frequency (SF) by an average of 20.11% and 23.63% respectively, compared to the SOTA method.

## 2 RELATED WORKS

### 2.1 Multi-Modality Image Fusion

In recent years, multiple deep learning-based methods have been developed to address the challenges posed by MMIF and the most commonly used networks are CNNs and GANs [2, 21, 24, 54, 81].

In CNN-based methods, various frameworks and loss functions are designed for feature extraction, feature fusion, and image reconstruction [27, 58, 74, 88]. Li *et al.* applied dense connections to extract features [26], and Wang *et al.* design multiple kernels to extract multi-scale features [67]. Besides, contrastive learning has been widely used to distinguish different modalities [33, 36, 90], and Liu *et al.* and Xu *et al.* perform image or feature decomposition before fusion [34, 75]. Moreover, multiple works combine CNN with transformers [29, 59, 60, 63]. For example, Ma *et al.* and Wang *et al.* utilize SwinTransformer to improve fusion performance [39, 66]. Additionally, some methods use prior knowledge of downstream tasks to assist in the loss function design. For example, Tang *et al.* [56, 58] use semantic segmentation masks, and Liu *et al.* [33] use salient masks to guide the training process.

GAN-based methods model the global information under unsupervised conditions [32, 84, 89]. Ma *et al.* first introduces GAN to fusion task [41]. Later, methods such as multi-classification GAN [42] and guided filters [78] are introduced. To balance each modality, asymmetric generator-discriminator structures are proposed [40, 85]. Moreover, Li *et al.* introduce the attention mechanism into the GAN-based fusion network [28].

### 2.2 Diffusion Model

Diffusion models have demonstrated powerful capabilities in various generation tasks [9, 18, 49, 52], including text-to-image [51, 82], image-to-image [46, 50, 62], image inpainting [1, 35], etc.

In addition, some works have explored the application of diffusion models, represented by DDPM [18], to high-level vision tasks, such as semantic segmentation [70] and object detection

**Figure 3: The overall framework of the proposed dual-branch DCAFuse (IVF as an example). Following our proposed timestep selection strategy, the diffusion-based branch models global information $F^D$ during the denoising process, while the CNN-based branch extracts local detailed features $F^C$. Subsequently, the proposed Complementary Feature Aggregation Module (CFAM) effectively aggregates them.**

[6, 14]. There are also some diffusion-based works focusing on low-level vision tasks, such as image restoration [37, 71], image super-resolution [64] and image fusion [30, 87]. In a framework for the above tasks, a commonly used method is first to introduce noise to source images, and then extract the latent features from the denoising U-Net for the following tasks [3, 4, 79]. Although diffusion models can produce visually appealing fused images, their limited local perception capabilities and inherent noise-adding mechanism result in significant detailed information loss.

## 3 METHOD

### 3.1 Overview

The proposed DCAFuse utilizes a dual-branch diffusion-CNN framework for comprehensive multi-modality image fusion. Taking the IVF task for instance, the RGB-channel visible image $X_{vis} \in \mathbb{R}^{h \times w \times 3}$ are combined with the infrared image $X_{ir} \in \mathbb{R}^{h \times w \times 1}$, forming the original input denoted as $X_0 \in \mathbb{R}^{h \times w \times 4}$.

As shown in Fig. 3, DCAFuse consists of diffusion-based and CNN-based branches. In the diffusion-based branch, we initially introduce noise into $X_0$ following our proposed timestep selection strategy, followed by intermediate feature extraction during the denoising process for global information modeling ($F^D$). In the CNN branch, multi-scale convolutional kernels and attention blocks are employed to extract and consolidate local detailed features ($F^C$). Subsequently, the Complementary Feature Aggregation Model (CFAM), a novel component of our approach, generates

coordinate-aware attention maps to capture the long-range dependencies between $F^D$ and $F^C$, allowing for effective aggregation. The aggregations are finally fed into the fusion head to obtain the fusion result.

### 3.2 Global Information Modeling

Through the denoising process, DDPM can encapsulate global information within intermediate features [20, 68]. In the diffusion-based branch, we first obtain noisy image $X_t$ for specified timestep $t$ by introducing Gaussian Noise, denoted as $\epsilon_t$, to $X_0$ then extract intermediate features from the denoising U-Net.

According to [18], instead of progressively adding noise, we can derive $X_t$ directly from a single operation as detailed below:

$$X_t = \sqrt{\bar{\alpha}_t} X_0 + \sqrt{1 - \bar{\alpha}_t} \epsilon, \tag{1}$$

where the noise $\epsilon \sim \mathcal{N}(0, I)$, and the variance $1 - \bar{\alpha}_t$ is related to the predefined variance schedule.

Subsequently, the noisy image $X_t$ is fed into the DDPM for a single-step denoising (reverse diffusion) process as follows:

$$X_{t-1} = \frac{1}{\sqrt{\alpha_t}} (X_t - \frac{1 - \alpha_t}{\sqrt{1 - \bar{\alpha}_t}} \epsilon_\theta(X_t, t)) + \sigma_t z, \tag{2}$$

where $z \sim \mathcal{N}(0, I)$, $\epsilon_\theta(X_t, t)$ represents the predicted noise, and $\sigma_t$ is related to the predefined variance schedule.

Eq. 1 and Eq. 2 are performed at $N$ timesteps (i.e. $t_1, t_2, \ldots, t_N$) to comprehensively capture the original information [3, 79]. Then,

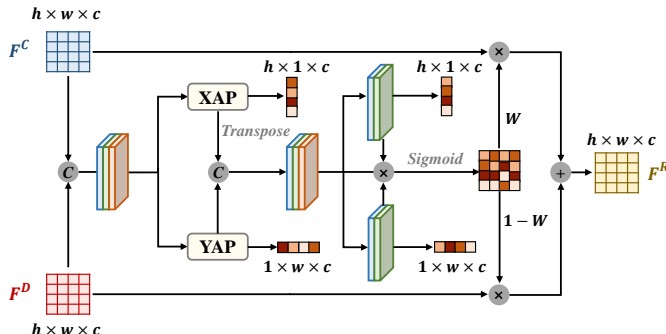

Figure 4: The proposed Complementary Feature Aggregation Module (CFAM). "XAP" and "YAP" represent average pooling along the X-axis (horizontal) and Y-axis (vertical) directions, respectively. $F^R$ denotes the aggregated features.

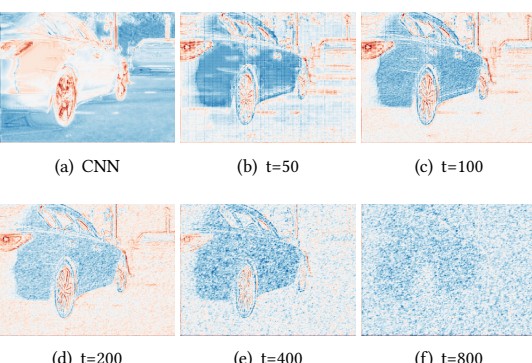

(a) CNN    (b) t=50    (c) t=100

(d) t=200    (e) t=400    (f) t=800

Figure 5: Visualization of features extracted from (a): CNN-based branch and (b) - (f): Block 14 of DDPM at different timesteps. Red represents stronger features, while Blue represents weaker features.

from $M$ distinct blocks in the denoising U-Net, we extract multi-scale intermediate features denoted as $F^D_{(i,j)}$, with $i \in \{1, 2, \ldots, M\}$, $j \in \{1, 2, \ldots, N\}$.

As depicted in Fig. 3, multi-timestep features extracted from Block $i$ (beginning from $i = 1$) are concatenated as $F^D_i$. Subsequently, the Cross-Timestep Feature Aggregator (CTFA) refines $F^D_i$ using a range of various convolutional and attention blocks. The refined feature is then upsampled to the same size as $F^D_{i+1}$, denoted as $F^D_{i'}$. Finally, $F^D_{i+1} = F^D_{i+1} + F^D_{i'}$ is obtained and then fed into the CTFA for the next iteration, continued until $i = M$. The concluding output from the diffusion-based branch is represented as $F^D$.

## 3.3 Local Detailed Feature Extraction

With superior local perception, CNN captures detailed features that serve as an effective supplement to the global information structured by DDPM [45, 55].

In the CNN-based branch, 3-stage convolutional layers along with Mixed Attention Blocks (MABs) are utilized for the extraction of multi-scale local detailed features, symbolized as $F^C_k$ where $k \in \{1, 2, 3\}$.

Subsequently, the Multi-Scale Feature Aggregator (MSFA) progressively merges $F^C_k$ [58]. Initially, $F^C_k$ is upscaled to match the size of $F^C_{k+1}$, following which the scaling factor $\gamma_k$ and bias $\beta_k$ are generated via MLPs to modulate $F^C_{k+1}$ as follows:

$$F^C_{k+1} = F^C_{k+1} \odot \gamma_k + \beta_k, \tag{3}$$

where $\odot$ denotes element-wise multiplication operation. Through Eq. 3, multi-scale local detailed features are fused into $F^C$.

## 3.4 Complementary Feature Aggregation Module

We design a novel Complementary Feature Aggregation Module (CFAM) to effectively aggregate the global information $F^D \in \mathbb{R}^{h \times w \times c}$ and local detailed features $F^C \in \mathbb{R}^{h \times w \times c}$.

Specifically, by generating the coordinate-aware attention maps of $F^{cat} = Concat(F^C, F^D) \in \mathbb{R}^{h \times w \times 2c}$, CFAM can capture its long-range dependencies in multiple directions, thus dynamically adjusting the aggregation weights.

Fig. 4 shows the specific workflow of the proposed CFAM. Initially, a $1 \times 1$ convolutional layer is utilized to adjust the number of channels (i.e. $2c \rightarrow c$). Then, CFAM extracts direction-aware feature maps $F^x \in \mathbb{R}^{h \times 1 \times c}$ and $F^y \in \mathbb{R}^{1 \times w \times c}$ by orthogonal 1-D average pooling layers as follows:

$$F^x, F^y = XAP(F^{cat}), YAP(F^{cat}), \tag{4}$$

where $XAP$ and $YAP$ represent performing average pooling along the horizontal and vertical directions, respectively. Given that $F^x$ and $F^y$ obtain the saliency information of features in corresponding directions, we concatenate them in the vertical direction and perform channel reduction by convolutional layer as:

$$F^{xy} = Concat((F^x)^T, F^y) \in \mathbb{R}^{1 \times (h+w) \times \frac{c}{r}}, \tag{5}$$

where $T$ denotes transposition operation, and $r$ represents the ratio of channel reduction. Afterward, by convolutional layers and nonlinear functions, $F^{xy}$ are encoded into 1-D coordinate-aware attention vectors $F^{CdA}_x \in \mathbb{R}^{h \times 1 \times c}$ and $F^{CdA}_y \in \mathbb{R}^{1 \times w \times c}$, which capture the long-range dependencies of input $F^{cat}$ along the corresponding spatial direction [19].

Subsequently, $F^{CdA}_x$ and $F^{CdA}_y$ are broadcast into $(H, W)$ and utilized to perform element-wise multiplication, resulting in coordinate-aware attention map $F^{CdA} \in \mathbb{R}^{h \times w \times c}$, which reflects long-range dependencies in all directions. Then CFAM aggregates $F^D$ and $F^C$ as follows:

$$F^R = F^{CdA} \odot F^D + (1 - F^{CdA}) \odot F^C, \tag{6}$$

where $F^R$ denotes the aggregated features. According to coordinate-aware attention map $F^{CdA}$, CFAM fully encapsulates the complementary attributes of dual-branch features, thus effectively aggregating the global information $F^D$ and local detailed features $F^C$. Finally, the aggregated features are fed into the fusion head to generate the MMIF results.

## 3.5 Timestep Selection Strategy

To discern the timestep selection strategy that effectively complements the denoising features and features extracted by the CNN-based branch, we examine latent feature representations across multiple timesteps. Fig. 5 (a) illustrates that the CNN-based branch focuses more on salient targets and local details while paying limited attention to background information.

As shown in Fig. 5(b), although existing methods generally select early timesteps (approximately $t = 50$) for feature extraction [3, 79], aiming to diminish noise-induced distortion of the original information, this approach fails to comprehensively portray the whole global scene. Fig. 5 (c)-(d) demonstrates that, with the progression of the sampling timestep, the diffusion model progressively captures background features, resulting in effective modeling of global information at $t = 200$. Nevertheless, when the timestep exceeds 800, the intense noise seriously precludes the extraction of information within the denoising process.

Overall, existing methods conventionally utilize early denoising timesteps for feature extraction, resulting in inadequate capture of global information. Stemming from our case analysis, we propose that features gathered at slightly late timesteps more effectively incorporate global information, serving as a suitable supplement to the local detailed features derived by CNN.

Following our proposed strategy, we execute ablation experiments, revealing $t = [100, 200, 400]$ as the relatively preferable denoising timesteps combination. Further specifics will be illustrated in the experiments.

## 3.6 Loss Function

The overall loss function consists of three components: intensity loss $L_{int}$, gradient loss $L_{grad}$, and the proposed cosine divergence loss $L_{CD}$. It can be formulated as:

$$L = L_{int} + \alpha L_{grad} + \beta L_{CD}, \qquad (7)$$

where $\alpha$ and $\beta$ denote the balancing factors of each loss term. Specifically, $L_{int}$ calculates pixel-wise intensity loss between the fused image $I_{fused}$ and input image $I_1, I_2$ (initial channel duplication will be applied to the single-channel input), which can be defined as:

$$L_{int} = \sum_{i=1}^{3} \|I_{fused}^i - max(I_1^i, I_2^i)\|_1. \qquad (8)$$

Similarly, $L_{grad}$ to calculate the gradient loss can be defined as:

$$L_{grad} = \sum_{i=1}^{3} \|\nabla I_{fused}^i - max(\nabla|I_1^i|, \nabla|I_2^i|)\|_1. \qquad (9)$$

Furthermore, we introduce a cosine divergence loss $L_{CD} \in [-1, 1]$, with 1 indicating complete similarity and -1 indicating absolute dissimilarity, which can be defined as:

$$L_{CD} = \frac{F^C \cdot F^D}{max(\|F^C\|_2, \epsilon) \cdot max(\|F^D\|_2, \epsilon)}, \qquad (10)$$

where $\cdot$ symbolizes the vector dot product and $\epsilon$ denotes a minimal constant to circumvent zero division. By minimizing the $L_{CD}$ between $F^C$ and $F^D$, DCAFuse is encouraged to better explore the complementary of these features, thus improving the fusion performance.

## 4 EXPERIMENTS

In this section, we carry out extensive experiments for infrared and visible image fusion (IVF) and medical image fusion (MIF) tasks. First, we introduce the experimental configurations and details. Subsequently, we undertake qualitative and quantitative comparisons of our proposed method with other state-of-the-art methods. Finally, various ablation studies are conducted to demonstrate the effectiveness of the proposed modules.

## 4.1 Setup

*4.1.1 Datasets.* The proposed DCAFusion is trained on the MSRS [57] training set (1083 pairs). For the IVF task, we choose three datasets for testing, i.e. RoadScene [73], MSRS [57] test set (361 pairs), and TNO [61]. As for the MIF task, we utilize three datasets collected by [86] from [44] for testing, specifically MRI-CT, MRI-PET, and MRI-SPECT. Notably, to measure the generalization performance, no additional datasets are incorporated for validation or fine-tuning.

*4.1.2 Metrics.* Six representative evaluation metrics are employed to evaluate the fusion performance of methods quantitatively, including standard deviation (SD) [47], entropy (EN) [48], visual information fidelity (VIF) [16], average gradient (AG) [7], edge information-based $Q^{AB/F}$ [76] and spatial frequency (SF) [11]. A higher score on these metrics indicates better fusion performance.

*4.1.3 Implement Details.* The total training process is divided into two stages: in stage 1, we train the DDPM for noise prediction in accordance with the training set described in [18]; in stage 2, we use denoising timesteps $t = [100, 200, 400]$ to extract intermediate features from the BLock $B = [2, 5, 8, 11, 14]$ of the frozen DDPM, which are subsequently utilized to train other components in DCAFusion. During the preprocessing stage, we crop images in the MSRS training set into patches sized 160×160 at random. In the training phase, we adopt the Adam optimizer with an initial learning rate of 0.0001 and set the batch size to 16. The balancing factors in loss function Eq. 7, namely $\alpha$ and $\beta$, are set to 1.00 and 0.05, respectively. All experiments are conducted on NVIDIA GeForce RTX 4090 GPUs and implemented on the PyTorch platform.

*4.1.4 Comparison Approaches.* We compare the proposed DCAFusion with seven state-of-the-art image fusion methods, including U2Fusion [72], SDNet [80], SwinFusion [39], TarDAL [32], CDDFuse [86], DDFM [87] and Dif-Fusion [79]. Methods using prior knowledge, such as [58] and [33], are not included in comparisons.

## 4.2 Infrared and Visible Image Fusion

On IVF datasets, we compare the fusion performance of DCAFusion with SOTA methods, qualitatively and quantitatively.

*4.2.1 Qualitative Comparisons.* As exhibited in Fig. 6, methods such as U2Fusion, SDNet, and DDFM appear to be under-exposed, causing the people in the red box to fade into the obscurity of the nighttime environment. In contrast, TarDAL tends to be over-exposed. Of all the methods, ours distinctly outlines the shapes of people and maximizes the traffic sign's contrast in the green box, improving its legibility. Moreover, as shown in Fig. 7, our approach

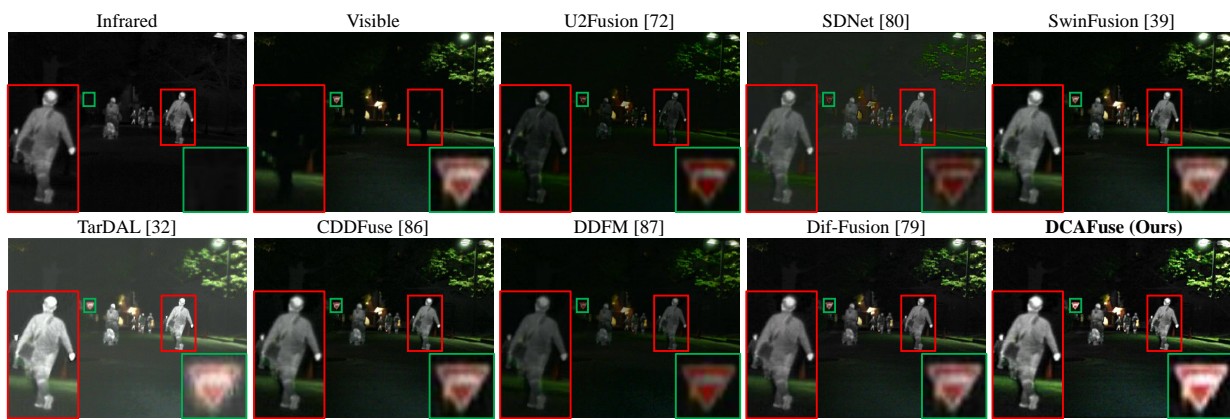

**Figure 6: Visual comparison of "00798N" on the MSRS IVF dataset [57].**

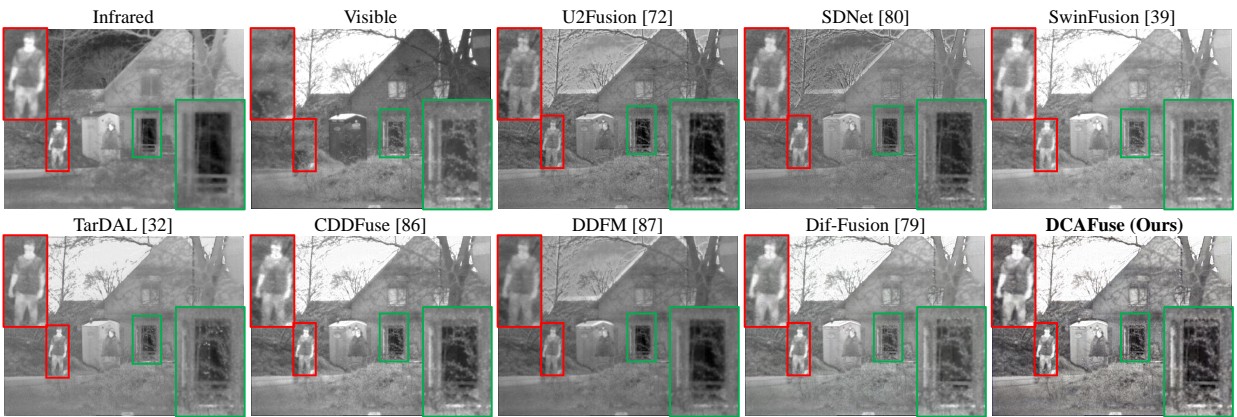

**Figure 7: Visual comparison of "meeting scene" on the TNO IVF dataset [61].**

**Table 1: Quantitative results of the IVF task on RoadScene [73], MSRS [57] and TNO [61] datasets. Red color and Blue color indicate the best and second-best results, respectively.**

| Methods | Dataset: RoadScene [73] | | | | | | Dataset: MSRS [57] | | | | | | Dataset: TNO [61] | | | | | |
|---------|-------|-------|-------|-------|-------------|-------|-------|-------|-------|-------|-------------|-------|-------|-------|-------|-------|-------------|-------|
| | SD | EN | VIF | AG | $Q^{AB/F}$ | SF | SD | EN | VIF | AG | $Q^{AB/F}$ | SF | SD | EN | VIF | AG | $Q^{AB/F}$ | SF |
| U2F [72] | 35.97 | 7.009 | 0.574 | 5.343 | 0.501 | 13.60 | 27.71 | 5.561 | 0.545 | 2.899 | 0.421 | 9.242 | 37.67 | 7.094 | 0.618 | 5.023 | 0.427 | 11.85 |
| SDN [80] | 40.33 | 7.136 | 0.595 | 5.612 | 0.500 | 14.28 | 17.32 | 5.255 | 0.489 | 2.707 | 0.370 | 8.691 | 33.78 | 6.695 | 0.577 | 4.630 | 0.427 | 11.64 |
| SwinF [39] | 45.32 | 7.053 | 0.672 | 4.345 | 0.498 | 11.68 | 42.98 | 6.622 | 0.990 | 3.564 | 0.642 | 11.08 | 39.39 | 6.881 | 0.749 | 4.195 | 0.521 | 10.64 |
| TarD [32] | 43.20 | 7.336 | 0.582 | 4.149 | 0.441 | 11.26 | 35.46 | 6.348 | 0.673 | 3.115 | 0.426 | 9.873 | 40.25 | 6.806 | 0.600 | 3.893 | 0.413 | 10.54 |
| CDDF [86] | 50.83 | 7.327 | 0.687 | 5.830 | 0.514 | 15.59 | 43.38 | 6.699 | 1.045 | 3.748 | **0.689** | 11.55 | 44.66 | 7.063 | 0.787 | 4.658 | 0.521 | 12.33 |
| DDFM [87] | 46.91 | 7.019 | 0.579 | 4.139 | 0.450 | 10.87 | 43.79 | 6.171 | 0.742 | 2.518 | 0.473 | 7.380 | 34.55 | 6.854 | 0.641 | 3.397 | 0.432 | 8.526 |
| DIF [79] | 44.22 | 7.142 | 0.588 | 5.518 | 0.516 | 14.06 | 41.90 | 6.660 | 0.827 | 3.889 | 0.583 | 11.63 | 38.77 | 6.916 | 0.597 | 4.306 | 0.465 | 10.77 |
| **Ours** | **53.71** | **7.348** | **0.716** | **7.082** | **0.563** | **19.06** | **52.42** | **6.929** | **1.062** | **5.245** | 0.621 | **16.01** | **48.75** | **7.217** | **0.836** | **6.164** | **0.523** | **16.23** |

retains the most intricate details, such as the textures of the door-frames and leaves in the green box, and also distinctly separates the thermal target (people) from the background. In summary, the proposed DCAFuse effectively combines the thermal saliency information from the infrared image with the detailed texture from the visible image, generating a fusion image with the finest visual effect.

*4.2.2 Quantitative Comparisons.* Table. 1 displays the quantitative comparisons using six evaluation metrics on the RoadScene, MSRS test set, and TNO datasets. Compared with state-of-the-art methods, our proposed DCAFuse stands out with superior performance. Specifically, achieving the best performance in SD and EN proves that our method is capable of integrating the richest original information. With $Q^{AB/F}$ maintaining a high level, our technique efficiently preserves edge contours. Achieving the best

**Table 2: Quantitative results of the MIF task on MRI-CT, MRI-PET and MRI-SPECT datasets [44]. Red color and Blue color indicate the best and second-best results, respectively.**

| Methods | Dataset: MRI-CT [44] | | | | | | Dataset: MRI-PET [44] | | | | | | Dataset: MRI-SPECT [44] | | | | | |
|---|---|---|---|---|---|---|---|---|---|---|---|---|---|---|---|---|---|---|
| | SD | EN | VIF | AG | $Q^{AB/F}$ | SF | SD | EN | VIF | AG | $Q^{AB/F}$ | SF | SD | EN | VIF | AG | $Q^{AB/F}$ | SF |
| U2F [72] | 55.36 | 4.883 | 0.364 | 6.459 | 0.477 | 23.12 | 53.35 | 4.330 | 0.438 | 5.607 | 0.435 | 19.23 | 46.52 | 3.912 | 0.454 | 4.036 | 0.513 | 15.96 |
| SDN [80] | 46.54 | 5.163 | 0.373 | 7.367 | 0.508 | 26.97 | 45.58 | 4.639 | 0.474 | 6.260 | 0.573 | 20.52 | 43.53 | 4.274 | 0.570 | 4.602 | 0.651 | 16.42 |
| SwinF [39] | 82.03 | 4.828 | 0.566 | 7.262 | 0.584 | 30.88 | 74.34 | 4.547 | 0.660 | 6.747 | 0.645 | 22.19 | 59.57 | 4.078 | 0.628 | 4.199 | 0.615 | 16.11 |
| TarD [32] | 59.37 | 5.202 | 0.453 | 5.054 | 0.342 | 19.33 | 57.63 | 4.695 | 0.568 | 5.248 | 0.481 | 18.82 | 51.49 | 4.336 | 0.455 | 3.839 | 0.443 | 16.34 |
| CDDF [86] | 81.39 | 4.711 | 0.499 | 7.880 | 0.596 | 33.90 | 74.36 | 4.196 | 0.649 | 6.883 | 0.644 | 24.62 | 60.20 | 3.857 | 0.599 | 4.320 | 0.640 | 17.13 |
| DDFM [87] | 59.91 | 4.528 | 0.449 | 5.031 | 0.415 | 20.68 | 61.22 | 3.917 | 0.652 | 5.325 | 0.552 | 18.87 | 58.27 | 3.802 | 0.611 | 3.684 | 0.608 | 14.42 |
| DIF [79] | 79.80 | 5.347 | 0.505 | 7.732 | 0.608 | 30.02 | 70.70 | **5.115** | 0.565 | 6.473 | 0.589 | 20.71 | 59.88 | 4.595 | 0.558 | 4.723 | 0.621 | 17.21 |
| **Ours** | **82.31** | **5.353** | **0.583** | **8.436** | **0.641** | **35.80** | **74.86** | 4.978 | **0.668** | **7.825** | **0.699** | **28.57** | **63.76** | **4.652** | **0.691** | **5.708** | **0.728** | **22.13** |

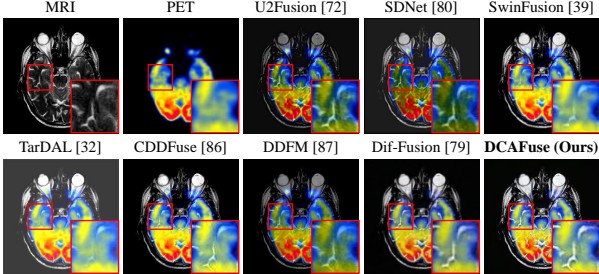

MRI   PET   U2Fusion [72]   SDNet [80]   SwinFusion [39]

TarDAL [32]   CDDFuse [86]   DDFM [87]   Dif-Fusion [79]   **DCAFuse (Ours)**

**Figure 8: Visual comparison on MRI-PET MIF dataset [44].**

VIF underscores that our method delivers the most appealing visual effects. Furthermore, the noticeable enhancements in AG and SF, by average increments of 26.36% and 30.52% respectively across the IVF datasets, validate that our results present the most detailed texture characteristics. Quantitative results prove that the proposed DCAFuse effectively integrates the saliency information in infrared images and the texture details in visible images.

## 4.3 Medical Image Fusion

In this section, we evaluate the fusion performance on MIF datasets without fine-tuning, aiming to assess the generalization performance of the methods.

*4.3.1 Qualitative Comparisons.* As demonstrated in Fig. 8, U2Fusion, SDNet, and DDFM are deficient in preserving the brightness information, leading to the distortion of significant color information originating from PET, while SwinFusion, TarDAL, CDDFuse, and Dif-Fusion tend to lose texture detail information from MRI, especially as emphasized in the red box. Serving as an exemplar, our proposed DCAFuse effectively leverages the abundant color information from PET while simultaneously maintaining distinct texture details from MRI, thus delivering the most appealing fusion effect.

*4.3.2 Quantitative Comparisons.* As illustrated in Table 2, the proposed DCAFuse yields the best or second-best performance across all metrics. Significantly, our method delivers average improvements of 13.87%, 16.74%, and 8.54% in AG, SF, and $Q^{AB/F}$ respectively, reflecting the capability of DCAFuse to exhibit the most distinct brain structures. Furthermore, DCAFuse posts outstanding scores in SD and EN, proving the full preservation of original information. Additionally, superior VIF underscores the fidelity

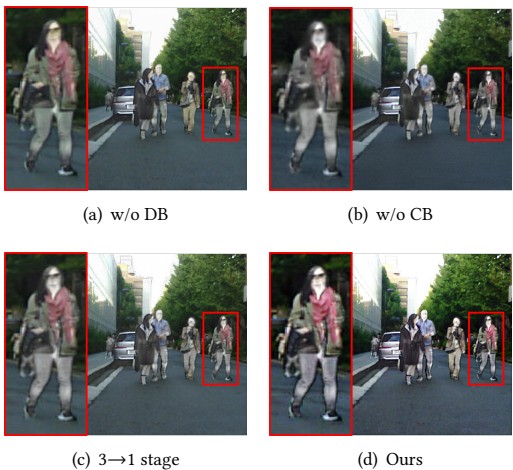

(a) w/o DB        (b) w/o CB

(c) 3→1 stage        (d) Ours

**Figure 9: Visual ablation comparisons of the framework. "DB" and "CB" denote the diffusion-based branch and the CNN-based branch correspondingly. "3→1 stage" indicates single-scale feature extraction in the CNN-based branch.**

of the visual information in our fused images, thereby providing effective assistance in medical diagnosis. Without fine-tuning, DCA-Fuse outperforms the state-of-the-art methods, demonstrating its remarkable generalization performance in diverse multi-modality image fusion tasks. Overall, quantitative evaluations demonstrate the superior performance of the proposed DCAFuse in integrating information procured from a myriad of medical imaging modalities.

## 4.4 Ablation Study

We conduct ablation experiments about the proposed dual-branch framework, CFAM, cosine divergence loss, and denoising timesteps, using the MSRS dataset for both training and testing procedures.

*4.4.1 Dual-branch Framework.* We first remove the diffusion-based branch and CNN-based branch independently, followed by substituting the 3-stage multi-scale feature extraction blocks with a single-scale block in the CNN-based branch. The qualitative and quantitative results are shown in Fig. 9 and Table. 3, respectively. As illustrated in Fig. 9(a), without the diffusion-based branch, the fused image loses saliency information from the infrared image, causing the person to lose the highlight. This matches the observed

**Table 3: Quantitative ablation results of the framework. "DB", "CB", and "3→1 stage" denote the diffusion-based branch, the CNN-based branch, and single-scale feature extraction correspondingly. Red color and Blue color indicate the best and second-best results, respectively.**

| Methods | SD | EN | VIF | AG | $Q^{AB/F}$ | SF |
|---|---|---|---|---|---|---|
| w/o DB | 48.38 | 6.792 | 1.009 | 5.097 | 0.612 | 15.52 |
| w/o CB | 41.77 | 6.854 | 0.820 | 3.813 | 0.582 | 11.39 |
| 3→1 stage | 42.05 | 6.665 | 1.027 | 3.840 | 0.613 | 11.69 |
| **Ours** | **52.42** | **6.929** | **1.062** | **5.245** | **0.621** | **16.01** |

**Table 4: Quantitative ablation results of the proposed CFAM. "CdA", "Conv", and "CSA" denote coordinate attention, convolution, and channel-spatial hybrid attention, respectively. Red color and Blue color indicate the best and second-best results, respectively.**

| Methods | SD | EN | VIF | AG | $Q^{AB/F}$ | SF |
|---|---|---|---|---|---|---|
| CdA→Conv | 45.48 | 6.788 | 1.041 | 4.572 | 0.566 | 13.80 |
| CdA→CSA | 41.74 | 6.804 | 0.595 | 3.611 | 0.446 | 11.15 |
| **Ours** | **52.42** | **6.929** | **1.062** | **5.245** | **0.621** | **16.01** |

**Table 5: Quantitative ablation results of the loss function. "$L_{NMSE}$" represents the Negative Mean Squared Error (NMSE) loss. Red color and Blue color indicate the best and second-best results, respectively.**

| Methods | SD | EN | VIF | AG | $Q^{AB/F}$ | SF |
|---|---|---|---|---|---|---|
| w/o $L_{CD}$ | 51.99 | 6.811 | 1.045 | 4.972 | 0.616 | 15.21 |
| $L_{CD}→L_{NMSE}$ | **55.26** | 6.853 | 0.847 | 5.117 | 0.502 | 15.78 |
| **Ours** | 52.42 | **6.929** | **1.062** | **5.245** | **0.621** | **16.01** |

deterioration in EN and $Q^{AB/F}$. Fig. 9(b) and Fig. 9(c) reflect that without a comprehensive CNN structure supplementing local information, the texture details of people and background appear blurry, corresponding to a notable decline in AG and SF. Our result, illustrated in Fig. 9(d), offers the most striking visual contrast and rich texture details.

*4.4.2  Complementary Feature Aggregation Module.* In the proposed CFAM, we substitute coordinate attention with convolution and channel-spatial hybrid attention, respectively. As shown in Table. 4, aggregation using convolution leads to a substantial decrease in EN, indicating a loss of original information. Besides, aggregation with channel-spatial hybrid attention results in a sharp drop in VIF and SF, indicating a deficiency in visual fidelity and texture detail. Departing from the above approaches, our proposed CFAM generates coordinate attention maps, seizing the long-range correlations of features both horizontally and vertically, thereby dynamically guiding the aggregation weights of branches. Quantitative results show that DCAFuse improves SD and AG by over 15.26% and 14.72%, demonstrating the effective incorporation of multi-modality information.

*4.4.3  Cosine Divergence Loss $L_{CD}$.* As shown in Table. 5, we initiate by omitting $L_{CD}$, which results in a decrease in EN, AG, and SF,

**Table 6: Quantitative ablation results of the denoising timesteps. "N/A" signifies non-convergence of the network. Red color and Blue color indicate the best and second-best results, respectively.**

| Timesteps | SD | EN | VIF | AG | $Q^{AB/F}$ | SF |
|---|---|---|---|---|---|---|
| 5, 25, 50 | | | N/A | | | |
| 5, 50, 100 | 49.41 | 6.851 | 1.050 | 4.927 | **0.644** | 15.02 |
| 50, 100, 200 | 50.81 | 6.876 | 1.051 | 5.032 | 0.637 | 15.47 |
| **100, 200, 400** | 52.42 | **6.929** | **1.062** | **5.245** | 0.621 | **16.01** |
| 200, 400, 800 | **53.55** | 6.920 | 1.060 | 5.145 | 0.621 | 15.81 |

indicating that the extracted features lack local detailed information. Further, we replace $L_{CD}$ with the Negative Mean Squared Error (NMSE) loss function, defined as $L_{NMSE} = -\frac{1}{n}\sum_{i=1}^{n}(F^C - F^D)^2$, where $F^C$ and $F^D$ denote the output features of CNN-based and diffusion-based branch, respectively. Although $L_{NMSE}$ amplifies the numerical difference between features, thereby boosting SD, it neglects the structural attributes of features, thus not performing optimally on other metrics. Our proposed $L_{CD}$ fosters feature complementarity by maximizing the cosine distance, leading to more comprehensive fusion results.

*4.4.4  Denoising Timesteps.* We establish five groups of denoising timesteps: earliest (5, 25, 50), slightly early (5, 50, 100), midterm (50, 100, 200), slightly late (100, 200, 400), and latest (200, 400, 800). Implementing the earliest timestep results in a failure of network convergence, as denoising U-Net cannot effectively comprehend the global information under extremely low-intensity noise. As demonstrated in Table. 6, enlarging the denoising timestep gradually improves the fusion effect. Upon setting the denoising timesteps to [100, 200, 400], DCAFuse yields the best performance in EN, VIF, AG, and SF, demonstrating the comprehensive preservation of both global information and local detailed features However, when the latest timesteps are employed, there's a decline in the fusion effect observed as a downturn in both SF and VIF, due to a significant reduction of the original information caused by excessive-intensity noise. The experimental results suggest that within a tolerable noise intensity range, a slight delay in denoising timesteps aids in enhancing the multi-modality image fusion effect.

## 5  CONCLUSION

In this paper, we introduce DCAFuse, a dual-branch Diffusion-CNN framework designed for multi-modality image fusion. We propose a novel complementary feature aggregation module, based on coordinate attention, to effectively integrate the global information extracted by the diffusion model and the local detailed features captured by CNN. Moreover, the complementarity of features extracted from the dual branches is further enhanced, benefiting from our introduced cosine divergence loss and timestep selection strategy. Extensive experiments on IVF and MIF datasets demonstrate that the proposed method achieves SOTA performance in multi-modality image fusion.

In the future, we aim to explore the potential of diffusion models to effectively model global information across a wider scope of image fusion tasks.

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
