# OpenReview forum: "DCAFuse: Dual-Branch Diffusion-CNN Complementary Feature Aggregation Network for Multi-Modality Image Fusion"
_acmmm.org/ACMMM/2024/Conference — MM2024 Poster_

### Official Review · Reviewer_KwyU · 2024-05-16

**Rating:** 3
**Confidence:** 4

**Summary:**

This paper proposes a dual-branch deep network model (named as DCAFuse) for multi-modality image fusion, which is built on DDPM and CNN and equipped with a complementary feature aggregation module. Experimental results on two fusion tasks are reported to demonstrate the effectiveness of the model.

**Strengths:**

The paper is well organized and easy to follow. The experimental evaluation is adequate.

**Limitations:**

1. The proposed model is a dual-branch framework, including CNN-based branch and diffusion-based branch. The CNN-based branch aims to extract local detailed features, while the diffusion-based branch is to capture the global information. However, in the diffusion-based branch, the output of DDPM is fed into the denoising U-Net. It is known that U-Net is a CNN, which is not competent to extract global information due to the limited receptive field of convolution kernels. It is unclear how to ensure that the diffusion-based branch based on U-Net can work as desired, i.e., capturing global information.

2. In Section 3.5, the paper claims “Fig. 5 (c)-(d) demonstrates that, with the progression of the sampling timestep, the diffusion model progressively captures background features, resulting in effective modeling of global information at 𝑡 = 200.” It is unclear to me why capturing background features can result in effective modeling of global information? How to define the global information?

3. One contribution of the paper is to introduce a cosine divergence loss, i.e., cosine similarity, which is defined by Eq. (10). However, this definition is inaccurate. Specifically, it measures the similarity between F^{C} and F^{D} by vector dot product. It is unclear how to compute the vector dot product of two tensors F^{C} and F^{D}.

**Suitability:**

3

---

### Official Review · Reviewer_ndGQ · 2024-05-28

**Rating:** 5
**Confidence:** 3

**Summary:**

This paper proposes a DCAFuse framework for multi-modality image fusion. It includes a novel complementary feature aggregation module, a cosine divergence loss function, and a unique denoising timestep selection strategy. Extensive experimental results show that the proposed DCAFuse outperforms other state-of-the-art methods in multiple image fusion tasks.

**Strengths:**

1) This paper introduces a novel view of the diffusion model (i.e., global information extractor).
2) The proposed fusion method does not need to perform too many iterations.
3) The experiments showed the promising results of the proposed method.

**Limitations:**

1）The Transformer is famous for its capability of modeling long-range dependencies. However, in the Related Work section, there appears to be no comparison between the diffusion model and the transformer. I recommend adding a discussion about the transformer to emphasize the advantages of the diffusion model for your specific task. Moreover, in your experiments, only one transformer-based method has been used for comparison.
2) About the Cross-Timestep Feature Aggregator, I suggest you provide a more detailed explanation and motivation. From Fig. 2, it may cause some misunderstandings, such as information redundancy.
3) In Fig. 5, I suggest you add more contents. MMIF is not a generative task. So I think it is better to add the original inputs and final results.

**Suitability:**

2

---

### Official Review · Reviewer_9UU7 · 2024-05-30

**Rating:** 3
**Confidence:** 2

**Summary:**

This paper presents a novel multi-modality image fusion (MMIF) method called Diffusion-CNN feature Aggregation Fusion (DCAFuse). MMIF aims to integrate complementary features from source images, such as target saliency and texture specifics, into a single fused image. Recent diffusion-based image fusion methods have shown promising results but suffer from reduced local feature perception and loss of original information due to their noise-adding mechanism.

**Strengths:**

Effective Feature Aggregation:
The DCAFuse method employs a novel Complementary Feature Aggregation Module (CFAM) that dynamically guides the fusion process by capturing long-range dependencies through coordinate attention maps. This enables the effective integration of global and local features, enhancing the overall quality and robustness of the fused images.

Dual-Branch Architecture:
The integration of a denoising diffusion probabilistic model (DDPM) and a convolutional neural network (CNN) allows DCAFuse to leverage the strengths of both global and local feature extraction. The DDPM-based branch excels at constructing global information, while the CNN-based branch focuses on capturing local details through multi-scale convolutional kernels, resulting in a more comprehensive and detailed fused image.

Enhanced Complementarity with Novel Loss Function:
DCAFuse introduces a novel loss function based on cosine similarity and a unique denoising timestep selection strategy. These enhancements improve the complementarity of features extracted from the dual branches, ensuring that the fused images maintain a high level of detail and accuracy, outperforming other state-of-the-art methods in multiple image fusion tasks.

**Limitations:**

1.Despite proposing the Cross-Temporal Feature Aggregator (CTFA) and Complementary Feature Aggregation Module (CFAM), the paper lacks an in-depth explanation of the physical significance of these modules. Specifically, the paper does not adequately explain the physical basis of CTFA in aggregating features across multiple time steps and how it effectively reduces the impact of noise on feature extraction. This lack of explanation undermines the credibility and interpretability of its technical approach.
2.The study lacks innovation in the field of multimodal image fusion. Although it adopts a dual-branch architecture combining diffusion models and CNNs, this combination has already been applied in existing literature. The paper fails to clearly demonstrate significant improvements or unique advantages of its method over existing technologies.
3.In the abstract, the paper highlights the Complementary Feature Aggregation Module (CFAM), but this module is not well showcased in Figure 3. Additionally, the aggregation method employed by CFAM does not seem to offer significant innovation compared to traditional gating mechanisms.

**Suitability:**

2

---

### Meta-Review · Area_Chair_gNNC · 2024-07-02

**Recommendation:** Accept (Poster)
**Confidence:** 4

**Metareview:**

Two of three reviewers tend to accept this paper. The main strengths include the novelty, extensive experiments, good paper organization.